# Selection of the Minimum Number of EEG Sensors to Guarantee Biometric Identification of Individuals

**DOI:** 10.3390/s23094239

**Published:** 2023-04-24

**Authors:** Jordan Ortega-Rodríguez, José Francisco Gómez-González, Ernesto Pereda

**Affiliations:** 1Department of Industrial Engineering, University of La Laguna, 38200 San Cristóbal de La Laguna, Spain; jortegar@ull.edu.es (J.O.-R.); eperdepa@ull.edu.es (E.P.); 2IACTEC Medical Technology Group, Instituto de Astrofísica de Canarias (IAC), 38320 San Cristóbal de La Laguna, Spain

**Keywords:** EEG, biometrics, brain–computer interface (BCI), support vector machine (SVM), phase locking value (PLV), asymmetry index (AI)

## Abstract

Biometric identification uses person recognition techniques based on the extraction of some of their physical or biological properties, which make it possible to characterize and differentiate one person from another and provide irreplaceable and critical information that is suitable for application in security systems. The extraction of information from the electrical biosignal of the human brain has received a great deal of attention in recent years. Analysis of EEG signals has been widely used over the last century in medicine and as a basis for brain–machine interfaces (BMIs). In addition, the application of EEG signals for biometric recognition has recently been demonstrated. In this context, EEG-based biometric systems are often considered in two different applications: identification (one-to-many classification) and authentication (one-to-one or true/false classification). In this article, we establish a methodology for selecting and reducing the minimum number of EEG sensors necessary to carry out effective biometric identification of individuals. Two methodologies were applied, one based on principal component analysis and the other on the Wilcoxon signed-rank test in order to reduce the number of electrodes. This allowed us to identify, according to the methodology used, the areas of the cerebral cortex that would allow selection of the minimum number of electrodes necessary for the identification of individuals. The methodologies were applied to two databases, one with 13 people with self-collected recordings using low-cost EEG equipment, EMOTIV EPOC+, and another publicly available database with recordings from 109 people provided by the PhysioNet BCI.

## 1. Introduction

The concept of biometrics, which comes from the words bio (life) and metrics (measurement), consists of those techniques for individual identification of people based on their physical or biological traits [1,2], which result in unique and irreplicable information. It is therefore currently the object of study for security systems. There are some commonly used biometric traits, such as fingerprints, DNA, or facial recognition, among others; however, over the last decade, it has been discovered that the analysis of electrical brain signals such as EMG or MEG (magnetoencephalography) as an alternative, is an extremely useful and unforgeable individual identification and recognition tool.

An interesting concept in this field is vitality detection, whose objective is an actual measurement of the biometric sample taken from the legitimate and living individual at the time and place of authentication. It improves the reliability of a biometric system by allowing the system to resist artefacts and ensure that no non-living or fake samples are accepted. Although current biometric systems use people’s physiological information for authentication, these measurements hardly detect their vitality. However, although they appear secure, it has been shown that a biometric system can be counterfeited with artefact samples; for example, the fingerprint system can be counterfeited with an artificial finger prepared from gelatin, silicon, latex, or Play-Doh [3].

There are currently numerous and diverse personal identification techniques through the analysis of biometric features of the neurophysiology of the human body, such as the eye retina, fingerprints, or facial recognition, among others [4]. Although today the use of common analysis processes for these traits is widespread, many of them require relatively high-cost (both economically and computationally) hardware and software systems. One of the most widespread biometric identifiers, used above all in forensic science, is the fingerprint, although there are currently techniques that allow its falsification, which is why the possibility of an individual’s information being stolen by falsifying their biometric features in some way that manages to deceive the security system in charge of protecting the said information is a fact that poses challenges to fields of research in this area [5,6].

In recent years, it has been proposed to measure the electrical activity of the superficial human brain for biometric recognition uses such as identification (one-to-many classification) and authentication (one-to-one classification) of individuals [7,8,9,10]. One way to measure the activity of the cerebral cortex noninvasively is the electroencephalogram (EEG). The EEG has been widely used in medicine [11,12] and non-medical applications, such as in the development of brain–machine interfaces (BMIs) [13,14]. Unlike other biometric measurements, EEG-based biometric measurements of an individual are difficult to falsify, and it can also be guaranteed that the identified person is alive. However, it should be noted that EEG signals exhibit the complexity that can be influenced by the passage of time due to various factors such as noise, drowsiness, changes in electrode conditions and the current mental state of each individual at any given moment [15]. This fact can make it difficult to obtain the same EEG signal twice for the same person, leading to inconsistencies in biometric identification. In this respect, biometric identification may be affected by the non-stationarity of the EEG recordings [16]. This inherent nature of the signal, caused by shifts in the covariance of power features, led to a decrease in the accuracy of the BCI model. To address this issue, proposals such as spatial filtering and stationarity subspace analysis (SSA) have been put forward to reduce non-stationarity [17]. Proper electrode placement is also critical in EEG recordings to eliminate difficulties related to feature covariance shifts. Changes in the covariance of EEG features due to different electrode positions can result in differences in the recorded signals, making it difficult to identify individuals accurately.

Concerning the selection of features that improve the performance of biometric identification systems, studies have recently been published suggesting that the subtraction of information related to functional connectivity between different regions of the brain is a feature that can potentially improve pattern classification tasks in EEG signals. Several studies have demonstrated that features derived from functional connectivity can significantly enhance the performance of EEG signal classification for biometric recognition systems [18,19]. Incorporating such features can improve the robustness of biometric recognition systems by considering the interdependence of EEG channels [20]. One of the most effective methods for achieving this objective includes the study of phase synchronization [21,22].

In this context, using an eyes-closed resting paradigm, Campisi et al. examined in 2011 the contribution to subject discrimination of different areas of the brain and frequency bands [23]. The results showed a decrease in classification accuracy when making use of information extracted from acquisition channels located over frontal brain areas compared to those located in occipital and temporal brain areas. This decrease was more pronounced when high-frequency EEG rhythms were filtered out. In particular, the best result was obtained with the T7–Cz–T8 electrode arrangement and a low-pass filter with a cut-off frequency of 33 Hz when an autoregression model was employed, and an accuracy rate of 96.08% was obtained with a database of 48 subjects.

There are a variety of analytical tools in the literature available for measuring the statistical interdependence between brain electrical signals. These are based on various mathematical principles that are implemented in time and frequency domains, capturing linear or nonlinear changes, such as correlation, phase coherence, and Granger causality [24,25,26,27,28]. These tools allow the estimation of information extracted from a phenomenon known as functional brain connectivity, which is related to the measurement of temporal communication values of neuronal activity between different areas of the brain that are anatomically separated [29,30]. In this regard, Daria la Rocca et al. proposed in 2014 a novel approach involving the extraction of information from functional connectivity on a database of 108 subjects in the resting state with eyes closed (REC) and eyes open (REO) recording paradigms [31,32].

More recently, in 2016, Douglas Rodrígues et al. presented a paper addressing the problem of reducing the number of acquisition sensors required while still being able to maintain competent performance [33]. In this case, a binary version of the Flower Pollination Algorithm with different transfer functions was evaluated to select the best subset of channels that maximizes accuracy. This optimization problem was carried out using the classifier. The experimental results obtained indicated that the proposed model was able to make use of less than half the number of sensors while maintaining recognition rates of up to 87%. In the same year, Toshiaki Koike-Akino et al. also analysed brain waves acquired from a commercial EEG device to investigate its user identification and authentication capabilities [34]. First, they showed the statistical significance of the P300 component in event-related potential (ERP) data acquired through 14 EEG channels on a sample of 25 subjects. They then analysed the application of a variety of machine learning techniques, making comparisons in the use of several of them in terms of subject identification performance, using dimensional reduction techniques on the signal samples before the classification stage. The experimental results of this study showed an identification accuracy of 72% when using a single 800 ms ERP. Furthermore, they showed that the biometric identification accuracy of individuals can be significantly improved to 96.7% accuracy by jointly classifying multiple epochs.

This work aims to achieve a ranking of those electrodes located on the surface of the scalp, and therefore of the corresponding brain regions, that provide the most relevant information for an EEG-based biometric recognition system. For this purpose, it is necessary to determine the order of relevance of the features extracted from the EEG recordings by applying principal component analysis and the Wilcoxon signed-rank test. Features extracted are the power spectrum, asymmetry index and information related to the functional connectivity of the brain by calculating the phase-lock value.

## 2. Materials and Methods

The steps followed for the identification of individuals using EEG-based biometric measurements are shown in Figure 1.

### 2.1. Experimental Procedure

The evaluation of the proposed method was conducted on two distinct datasets. Dataset I was the primary dataset and consisted of a self-collected dataset using an inexpensive EEG acquisition device. Moreover, the largest sized available EEG dataset in the related literature, which we will refer to as dataset II, was utilized to contribute a greater degree of robustness to the validation of the obtained results. Dataset II was a publicly available collection of EEG recordings provided by the PhysioNet BCI [35]. The signal acquisition protocol was similar in both cases.

In the case of dataset I, EEG signals from thirteen volunteer healthy right-handed subjects (aged 18–51 years) with no motor pathology were recorded in a typical office environment. The experimental procedure has been described by Ortega et al. [7].

Seated in front of a computer screen with a black background on which instructions to be followed are displayed, each volunteer performed a specific mental task. This task consisted of performing a motor imagery action of squeezing a flexible object with the right hand [36]. For each subject, EEG signals were recorded in the basal state for 20 s and a 10 s transition period; finally, the main action (motor imagery action) was performed for another 20 s. EEG recordings were obtained from each participant in a single session of four repetitions, resulting in a total recording time of 80 s per participant. The study was approved by the ethical committee of the University of La Laguna (registration number: CEIBA2020-0405).

Dataset II consisted of EEG recordings from 109 healthy individuals obtained from the publicly available PhysioNet BCI database. In that case, each participant performed four different mental tasks involving eye, fist, and foot movements, with each task being repeated three times for a duration of two minutes per recording. During the tasks, a target was presented on the monitor’s right or left side to cue the participants to perform the corresponding action until the target disappeared. The EEG recordings of the motor imagery tasks and the sections involving right-hand movements were selected from the dataset. Three two-minute EEG recordings per participant were chosen for a total of 327 EEG records.

### 2.2. Data Acquisition

EEG recordings in dataset I were performed with the Emotiv Epoc+ (Emotiv Inc., San Francisco, CA, USA) portable commercial electroencephalograph. This device was suggested in [26] as the best low-cost EEG device in terms of versatility. This has 14 sensor electrodes (AF3, F7, F3, FC5, T7, P7, O1, O2, P8, T8, FC6, F4, F8 and AF4) with saline-soaked felt pads. The electrical reference point CMS (Common Mode Sense) is located at P3 or right mastoid (active electrode), and the noise cancellation electrode DRL (driven right leg) is located at P4 or left mastoid (passive electrode). The electrodes were placed on the scalp following the international 10–20 system. The EEG signals were transmitted to the computer with a 128 Hz sampling frequency using a wireless Bluetooth dongle and stored in the European Data Format (EDF).

Dataset II was built with EEG recordings using the BCI2000 (Laboratory of Nervous System Disorders, Wadsworth Center, New York State Department of Health, Albany, NY, USA) as the brain–computer interface system [37] with 64 electrodes. In that case, the electrodes were placed in agreement with the international 10–10 system. The EEG signals were collected with a 160 Hz sampling frequency and stored in EDF+ format.

### 2.3. Data Preprocessing

Preprocessing of the EEG signals was performed using FieldTrip, version 20230118 [38], a freely available toolbox for MATLAB^®^, The MathWorks, Inc. The chosen environment for this study was Matlab R2022b. The first step consisted of applying a baseline correction based on the mean average voltage to the EEG signals. Next, the signals were filtered using a bandpass filter (FIR) from 5 to 40 Hz to reduce noise. Finally, FieldTrip functions specifically designed to remove other artefacts, such as eye movements and muscle activation, were used [7,39].

Each EEG recording was segmented into epochs of 2 s duration and filtered into the beta frequency band (13–30 Hz). According to the related literature, the beta frequency band is considered to provide particularly valuable information in mental tasks involving motor imagery and motor action. Therefore, this frequency band should be carefully considered in any biometric identification application that involves such mental tasks [40].

### 2.4. Feature Extraction

For the proposed biometric identification model, three sets of features were extracted independently from each epoch of EEG signals in the beta frequency band: power spectrum, asymmetry index, and phase-locking value.

The power spectrum (PS) was calculated with the fast Fourier transform using multiple tappers from discrete spheroidal sequences [41].

The asymmetry index (AI) [42] was calculated as the Napierian logarithm of the fraction between the spectral power values of the signal acquired by the corresponding pairs of electrodes of each cerebral hemisphere (PS_chleft_, PS_chright_).
(1)$$Asymmetry\;Idex=ln\frac{{PS}_{chleft}}{{PS}_{chright}}$$

Information related to the functional connectivity of the brain was extracted by calculating the phase lock value (PLV) using the implementation described in [43].

For a given number of epochs (*N*) and a difference between the instantaneous phase of the two EEG signals, *θ* (*t*, *n*), at a specific time (*t*) and epoch (*n*), the PLV is defined as
(2)$$PLV\left(t\right)=\frac{1}{N}\left|\sum_{n=1}^{N}e^{i\theta \left(t,n\right)}\right|$$

This calculation describes the average of the absolute values of the phase difference between the two signals for each epoch and can identify transient phase lock values independently of the signal amplitude [44].

Hence, in the case of dataset I, every subject was represented by 112 features for every epoch in the beta frequency band. These features included 14 PS features, 7 AI features, and 91 PLV features. Similarly, for dataset II, each subject was characterized by 2107 features from the beta frequency band, comprising 64 PS features, 27 AI features, and 2016 PLV features. The dimension of the feature tables, which includes the subject’s label column, was 520 × 112 and 2289 × 2108 for dataset I and II, respectively, when using all available electrodes.

### 2.5. Feature Selection

In this study, two different techniques were independently used and compared as feature selection methods: principal component analysis (PCA) and the Wilcoxon signed-rank test.

Principal component analysis (PCA) is a statistical technique that allows the complexity of sample spaces with high spatial dimensionality to be reduced while preserving their principal information [45,46,47,48]. PCA can then be used to identify the most important features in the dataset and reduce their dimensionality while preserving the most important information. It is an unsupervised learning algorithm that studies the relationship between the variables that make up a data set to identify subgroups in which the data variation is maximum. To do this, it performs the calculation of geometric projections of the source data on lower-dimensional directing predictors called principal components (PCs), which are linear combinations of the original variables. The basic idea behind PCA is therefore to find a new set of variables (PC) that capture the most important information in the data. These new variables are linear combinations of the original variables and are chosen such that they are uncorrelated and ordered by the amount of variance they explain in the data.

The Wilcoxon signed-rank test is defined as a non-parametric statistical test that allows determining the correlation between variables of a pair of independent samples that do not follow a normal distribution; that is, between two distinct sets of items where the values of one sample do not reveal information about the values of the other [49]. Through this test, it is possible to determine the *p* value between the measurable characteristics on different data sets that allow the computation of the degree of correlation between them. Thus, it is determined which characteristics stand out for providing more decisive information.

In this context, each method was independently used and evaluated for the discovery and ordering of those variable characteristics of a data set that provide more significant information, given that they present a lower degree of correlation than the rest.

These two techniques are widely used in problems of dimensionality reduction in multivariate models through the selection of characteristics to simplify their complexity. Restrictive dimensionality reduction methods are of great interest in problems such as the selection of features with the highest extractable significance from an EEG for various applications, including the biometric identification of individuals. However, beyond this traditional use, they not only allow the simplification of the model but also facilitate the selection of those data acquisition channels that appear more frequently in these selected features and, consequently, make it possible to study the selection of the most relevant brain regions for each EEG application.

### 2.6. Channel Selection

After applying the proposed feature selection techniques, the EEG channels most commonly used by these selected features were determined. The goal was then to reduce the number of EEG channels required to achieve the desired level of accuracy. Accordingly, the most relevant channels were analysed in the self-collected dataset I. Therefore, the superficial regions of the cerebral cortex corresponding to the location of these favourite electrodes located on the scalp were sorted according to the relevance contained in the information provided by each one.

From the proposed feature selection methods, the corresponding reduced arrays were computed in which the features of each group—PS, AI or PLV—were ordered from highest to lowest relevance and, consequently, the most important channels to characterize the different EEG signals of each individual. For this purpose, the order of each one was established by assigning a corresponding score calculated as follows:(3)$${Score}_{ch}=\sum_{i=1}^{m_{i}}W_{{ch}_{ij}},$$
where
(4)$$W_{{ch}_{ij}}=k_{j}\frac{P_{i}}{f_{j}}$$

Considering $W_{{ch}_{ij}}$ as the weight value of each EEG channel (ch) as a function of a dimensionless constant *k_j_*, whose value in each case was established with the contribution of its corresponding feature group (PS, AI or PLV) by itself to the classification accuracy, *P_i_* is the number of ordered positions occupied by each feature in the feature array by PCA or the Wilcoxon test, *f_j_* is the total number of features in each corresponding group and *m_i_* is the total number of weights for each channel *ch_i_* of the EEG. The selected values of *k_j_* for the different feature groups were *k*_PS_ = 2 for spectral powers, *k*_AI_ = 1 for asymmetry indices and *k*_PLV_ = 1.2 for phase-locked values.

The PCA algorithm assigned a certain weight value to each feature in the different principal components. For each different feature, the value of its corresponding weight in all the main components extracted has been extracted to establish an order of relevance. After applying Equations (3) and (4) considering the sorted features that make up the first principal component of each group, the EEG channels were sorted from the highest to the lowest obtained score.

When using the Wilcoxon signed-rank test to sort the channels by the *p*-value of their characteristics involved, a reduced array of selected features was extracted. Unlike the PCA method, the Wilcoxon test can be performed only on paired data sets; that is, between the data of only two people at a time. For this reason, it was necessary to perform the test on 77 combinations of pairs of subjects for 13 subjects. Considering all the tests that sorted the features according to their *p*-value and the corresponding group of features, they were reordered by their statistical mode or the number of times they were repeated in the features array ordered by the Wilcoxon test among all the tests.

A plot of the method comparison between the PCA and the Wilcoxon test is depicted in Figure 2. The evaluation of the use of PCA or Wilcoxon signed-rank test for EEG channel selection was made by determining which of the two techniques provided a channel ranking that gave the best classification accuracy for biometric identification while using the least number of channels in dataset I (13 subjects). Finally, the corresponding extracted surface regions of the cerebral cortex were also evaluated on dataset II (109 subjects).

### 2.7. Classification. Support Vector Machine

In the present study, Support Vector Machines (SVM), from the MATLAB Classification Learner toolbox (MATLAB^®^, MathWorks, Inc., Portola Valley, CA, USA) with a Gaussian Radial Basis Function (RBF) kernel, were used as the classification algorithm. SVM is one of the most widely used algorithms owing to its simplicity and the excellent results it has provided [50,51,52].

An RBF kernel SVM is a type of SVM that uses a nonlinear kernel function to map the input data into a higher-dimensional space where it becomes linearly separable by a hyperplane. The solution of the classification problem is
(5)$$f\left(x\right)=C\left(\sum_{i}^{N}{\alpha_{i}y_{i}k(x}_{i},y_{i})+b\right)$$
where *x_i_* is the input vector ($x\in R^{N}$), *y_i_* is the class label (*y* ∈ {−1, +1}), *α_i_* is a set of Lagrange multipliers needed to solve the constrained optimization problem ($0\leq \alpha_{i}\leq C$, *C* is the box constraint), b stands for the bias and *k* is the RBF kernel defined as the exponential of the squared distance between two points in the feature space, which can be given by
(6)$$k\left(x_{i},y_{i}\right)=e^{\left(-\gamma {\left\|x_{i}{-y}_{i}\right\|}^{2}\right)}$$
where *γ* is the kernel scale. The parameters *C* and *γ* were fit to have the best classification precision.

The feature tables extracted from EEG dataset I and II, which contained the PS, AI, and PLV features measured in the beta frequency band, were split into training and validation sets. This split was performed using a ten-fold cross-validation technique, which is commonly used in machine learning to evaluate the performance of a model while preventing overfitting. In cross-validation, the data is divided into *k* equal parts, and the model is trained *k* times, with each part serving as the validation set once. The average performance across all the *k*-folds is then reported as the overall performance of the model. This approach helps to ensure that the model generalizes well to unseen data and is not overly influenced by noise or outliers in the training set. In this way, the inputs of the classifier are the feature data and the output is the label of the corresponding subject.

### 2.8. Computation Setup

The calculations involved in the present study were performed on a computer with an AMD Ryzen 7 3800X (Advanced Micro Devices, Inc., Santa Clara, CA, USA) processor with 8 cores and 16 threads at 4.5 GHz, an Nvidia RTX 2060 (NVIDIA Corp., Santa Clara, CA, USA) graphics card with 6 Gb of memory at 1.7 GHz, and four 4 × 16 Gb (64 Gb) RAM modules with a CAS latency of 16 at 3.2 GHz.

## 3. Results

In this section, we provide the classification performance obtained from the two proposed feature selection methods (PCA and Wilcoxon signed-rank test) applied to EEG datasets I and II. The effectiveness of the selected methods was evaluated in terms of reducing the number of EEG channels to a minimum while maintaining a desirable level of precision in the biometric identification system. The cerebral cortex regions corresponding to the selected electrodes were determined based on their locations that guaranteed accurate biometric identification in dataset I, using each of the proposed methods. These regions were then evaluated in dataset II using the corresponding electrodes that are approximately located within those regions. The results obtained using the proposed methods are presented in Table 1, Table 2, Table 3, Table 4, Table 5, Table 6, Table 7 and Table 8. As an initial benchmark, Table 3 shows that by using all 14 available electrodes in dataset I, 100% of classification accuracy was achieved to identify the 13 participants. Similarly, in the case of dataset II, Table 4 shows that 99.9% accuracy was achieved by utilizing all the 64 available electrodes to identify 109 participants.

### 3.1. Channel Selection Using PCA

After applying the PCA method to the feature table extracted from dataset I, the first component of each group of features was selected to carry out the ordering of the acquisition channels based on the score obtained. In Figure 3, the ratio of conservation of original data is represented according to the number of main components extracted for the groups of features PS, AI and PLV, correspondingly. The results showed that to keep 90% of the original information, it was necessary to extract three principal components for the PS set, eight for the AI set, and 48 for the PLV set. In other words, when using only the first principal component, the percentages of conservation of the original information on the different groups of characteristics were 71% for PS, 54% for AI and 39% for PLV.

Table 1 shows an example of calculating the weights $W_{{ch}_{ij}}$ and the score for the case of the T8 channel after using the PCA method for feature ordering. Likewise, Table 2 shows the order obtained from all the channels based on the score calculated after having used the PCA on each group of features.

Once the EEG acquisition channels are sorted by relevance, the performance of the Gaussian kernel SVM classifier is shown in Table 3. The metric results of precision, sensitivity (recall), F1-score, and MCC (Mathew’s correlation coefficient), as well as the number of features used as a function of the number of acquisition channels chosen by the PCA method, are shown. The precision results for biometric identification were maintained with values above 82% using a minimum of three channels and above 90% using a minimum of four channels. The computation time for the PCA method was 36.2 ms. From these results, it stands out that when features were selected in the training of the classifiers that use three channels or less, their performance decreased drastically.

The representation of the evolution of the degree of precision as a function of the number of channels shown in Figure 4a reveals a clear and expected trend towards better performance, in terms of higher accuracy values and lower standard deviations, as the number of EEG channels used increases.
sensors-23-04239-t001_Table 1Table 1Example of calculation of weights and scores of an EEG channel for its ordering after using the PCA method.PiFeaturesPS (kPS = 2)
AI (kAI = 1)
PLV (kPLV = 1.2)
91

O1–T8…

…14AF3
F7–T713F8
P8–AF412FC6
F3–O211T8
T7–P710T7
F3–P89AF4
AF3–AF48FC5
AF3–F37F7AF3–AF4F7–AF46F4F3–F4F3–P75P7T7–T8F3–F44F3FC5–FC6F3–T73O2O1–O2F4–AF42O1F7–F8F3–AF41P8P7–P8F3–FC5Weight for T8{WT8PS1=21114}{WT8AI1=157}{WT8PLV1=1.29191, WT8PLV2=1.28691, …, WT8PLV67=1.22491}Score for T8WT8PS1WT8AI1+WT8PLV1+WT8PLV2+…+WT8PLV67=13.61
sensors-23-04239-t002_Table 2Table 2EEG channels ordered by relevance using the principal component analysis method.OrderChannelScore1T813.612F813.523FC612.394FC511.685O111.406O210.677F410.28F78.999AF38.5310P78.5111T78.3812P87.1813AF45.3514F33.68
sensors-23-04239-t003_Table 3Table 3Results of classification metrics based on the number of EEG channels used through the application of PCA.Set of ChannelsSet of FeaturesPrecision (%)Recall (%)F1-Score (%)MCC (%)14112100100100100139899.62 ± 1.3999.62 ± 0.9299.61 ± 0.8099.58 ± 0.86128399.42 ± 1.5099.43 ± 1.0899.42 ± 0.9899.38 ± 1.05117099.04 ± 1.6399.09 ± 2.0499.04 ± 1.1498.96 ± 1.22105898.65 ± 2.4298.66 ± 1.9298.65 ± 1.9698.54 ± 2.1194897.88 ± 2.0097.93 ± 2.2097.89 ± 1.5697.71 ± 1.6883997.12 ± 2.4797.23 ± 3.2797.13 ± 2.0296.88 ± 2.1773095.77 ± 2.7795.90 ± 3.4595.78 ± 2.1395.43 ± 2.2862395.00 ± 3.8295.07 ± 3.9495.00 ± 3.4194.59 ± 3.6951691.73 ± 8.1391.88 ± 6.4491.67 ± 6.4691.07 ± 6.9341190.00 ± 11.6889.93 ± 8.1089.76 ± 9.4989.20 ± 10.003682.88 ± 13.0282.86 ± 10.4282.71 ± 11.1881.49 ± 12.062366.73 ± 21.5268.35 ± 20.1466.55 ± 19.8564.11 ± 21.121140.38 ± 28.9546.91 ± 29.5336.00 ± 22.6435.91 ± 20.57


Figure 5 shows the topographical maps estimating the relevance of different cerebral cortex areas underlying the surface location of the channels selected by PCA, considering their corresponding classification precision result. As the number of electrodes used decreases, there is a trend for their concentration in the locations close to the parietal and right temporal lobes.

To further study the relevance of these specific cerebral cortex areas, the locations of the four electrodes (T8, F8, FC6, and FC5) selected by PCA that guaranteed an accuracy rate above 90% in dataset I were compared to corresponding channels in dataset II. These channels included AF8, F8, F6, F4, FT8, FC6, FC4, T8, CP6, C4, C6, TP8, FC5, FC3, C5, and C3. Table 4 complements these findings by reporting the classification metrics results of biometric identification achieved with dataset I and II using the corresponding set of channels located in the above mentioned cerebral cortex regions of interest in Figure 5. In addition, the results obtained when using all available channels in dataset II (64 channels) are also presented for comparison.
sensors-23-04239-t004_Table 4Table 4Classification results for dataset I and II by using the channels located on the cerebral cortex regions of interest extracted by PCA.DatasetSetof ChannelsPrecision (%)Recall (%)F1-Score (%)MCC (%)I (13 subj.)490.00 ± 11.6889.93 ± 8.1089.76 ± 9.4989.20 ± 10.00II (109 subj.)1699.22 ± 1.8499.20 ± 2.0299.19 ± 1.5199.18 ± 1.51II (109 subj.)6499.91 ± 0.6499.92 ± 0.6199.91 ± 0.4599.91 ± 0.52 


### 3.2. Channel Selection by Wilcoxon Signed-Rank Test

In this section, the results obtained by using the Wilcoxon signed-rank test as a method for ordering features from the table of features of the database I and the corresponding order of electrodes are shown. Table 5 illustrates an example of calculating the weights and score for the case of the AF4 channel, and Table 6 shows the order of the channels according to their calculated score. The computation time for the Wilcoxon signed-rank test method was 7.37 ms.

Once the EEG channels have been sorted, Table 7 displays the corresponding results of the classification metrics. In this case, as with the PCA method, high classification precision values are obtained using more than three channels with 80% and greater than 90% using four channels, although with a higher standard deviation and worse performance when the number of channels used is between 1 and 2 than in the case of the PCA method.

In the case of using the Wilcoxon test method, the representation of the evolution of the degree of accuracy as a function of the number of channels shown in Figure 4b also reveals, as occurred with the application of the PCA method, a clear and expected trend towards better performance, in terms of higher accuracy value and lower standard deviation, as the number of EEG channels used increases.

Figure 6 shows a graphical representation of the topographic evolution of precision, showing a tendency to concentrate them in the areas closest to the left frontal lobe as the number of electrodes used decreases.

The locations of the four electrodes (AF3, F3, F7, and FC5) selected by the Wilcoxon signed-rank test, which ensured an accuracy rate of over 90% in dataset I, were compared to the corresponding channels in dataset II that approximately match the corresponding location of those regions of the cerebral cortex. The considered channels for that case were F7, F5, F3, F1, Fz, AF7, AF3, AFz, FP1, FPz, FT7, FC5, FC3, FC1, C5, and C3. Table 8 displays the classification metrics results of biometric identification achieved with dataset I and II using the corresponding set of channels located in the left frontal lobe, as shown in Figure 6.
sensors-23-04239-t005_Table 5Table 5Example of the calculation of weights and scores of an EEG channel for its ordering after using the Wilcoxon signed-rank test method.PiFeaturesPS (kPS = 2)
AI (kAI = 1)
PLV (kPLV = 1.2)
91

AF–3F3…

…14

F3–P813

T7–F812AF3
AF3–O111F3
F7–AF410FC6
F7–FC69AF4
F3–FC58AF4AF3–AF4P7–AF47FC6F7–F8T7–T86F8F3–F4P7–F85AF4FC5–FC6P8–AF44O2T7–T8F7–FC63O2T7–T8F7–O22T7O1–O2F7–O11O1O1–O2P7–T8Weight for AF4{WAF4PS1=2912, WAF4PS2=2812, …, WAF4PS12=2512};{WAF4AI1=188};{WAF4PLV1=1.28191, WAF4PLV2=1.28691, …, WAF4PLV95=1.2591}Score for AF4WAF4PS1+…+WAF4PS12+WAF4AI1+WAF4PLV1+WAF4PLV2+…+WAF4PLV95=9.74
sensors-23-04239-t006_Table 6Table 6EEG channels ordered by relevance using the Wilcoxon signed-rank test method.OrderChannelScore1AF314.582F313.793F711.504FC511.075T710.206F49.747AF49.648P79.559T88.2410FC68.1011F88.0312O26.9013O16.0114P84.58
sensors-23-04239-t007_Table 7Table 7Results of classification metrics based on the number of EEG channels used through the application of the Wilcoxon signed-rank test method.Set of ChannelsSet of FeaturesPrecision (%)Recall (%)F1-Score (%)MCC (%)14112100100100100139899.51 ± 1.0899.44 ± 1.0799.42 ± 0.6599.38 ± 0.70128399.40 ± 1.4399.44 ± 1.0799.42 ± 0.8499.38 ± 0.89117099.23 ± 1.5899.26 ± 1.5199.23 ± 0.9699.17 ± 1.03105899.23 ± 1.2099.24 ± 1.1899.23 ± 0.8199.17 ± 0.8894899.23 ± 1.2299.25 ± 1.5299.23 ± 1.0899.17 ± 1.1783997.69 ± 2.4797.73 ± 2.3197.69 ± 1.7597.50 ± 1.8973096.92 ±4.1097.01 ± 3.6796.92 ± 3.2296.67 ± 3.4762395.19 ±4.7395.31 ± 4.7895.20 ± 4.2094.80 ± 4.5251692.31 ± 5.5492.56 ± 7.0492.36 ± 5.6691.68 ± 6.1441190.58 ±6.0590.81 ± 6.7190.60 ± 5.6989.81 ± 6.153680.58 ± 11.0280.88 ± 11.1080.57 ± 10.4878.98 ± 11.332366.38 ± 18.5159.63 ± 15.8859.55 ± 16.9357.17 ± 17.711142.31 ± 26.1138.83 ± 24.8838.63 ± 23.6637.98 ± 23.25
sensors-23-04239-t008_Table 8Table 8Classification results for dataset I and II by using the channels located on the cerebral cortex regions of interest extracted by Wilcoxon signed-rank test.DatasetSetof ChannelsPrecision (%)Recall (%)F1-Score (%)MCC (%)I (13 subj.)490.58 ± 6.0590.81 ± 6.7190.60 ± 5.6989.81 ± 6.15II (109 subj.)1699.52 ± 1.4199.51 ± 1.4599.51 ± 1.1299.50 ± 1.02II (109 subj.)6499.91 ± 0.6499.92 ± 0.6199.91 ± 0.4599.91 ± 0.52


## 4. Discussion

In the present study, we proposed a method for the calculation and establishment of an ordering of EEG channels based on the degree of relevance provided by the features in which they are involved. The applied feature ordering techniques, namely, PCA and the Wilcoxon signed-rank test, were independently compared for this purpose. Based on the use of both techniques, the obtained results indicated that a minimum of four EEG electrodes is recommended to achieve sufficient biometric identification accuracy on the self-collected dataset of 13 individuals using a non-expensive EEG device. This resulted in a 75% reduction with respect to the initial used quantity of electrodes, from 16 to 4, by selecting the electrodes located in the corresponding cerebral cortex regions of interest.

Regarding the location of the selected channels, as the number of electrodes used in both feature selection cases (PCA or Wilcoxon test) was reduced, the locations of the electrodes were concentrated in opposite lobe hemisphere areas. In this context, these extracted regions of interest were different depending on the employed feature selection method. We have found that when using PCA, the electrode placement concentrated on the right parietal and temporal lobes, whereas in the case of using Wilcoxon signed-rank test, the regions of interest were found near the left frontal lobe. Furthermore, the evaluation of biometric identification performance on a dataset of 109 individuals, considering the influence of these regions, revealed that the classification accuracy remained desirable and consistent, even with a significant increase in the number of subjects. The reduction in electrode usage was also 75%, as in the previous case, where the 64 initial electrodes were reduced to 16, which were localized in the aforementioned cerebral cortex areas of interest.

By contrast to alternative approaches, some of the most recent studies in the related literature have employed various electrode selection techniques for the problem of biometric identification of individuals based on EEG. Among these techniques, those related to the use of genetic algorithms (GA) stand out as prominent, as they can search a large search space to find the best possible solution to the problem.

In this context, a genetic algorithm-based method for reducing the number of EEG electrodes to those providing maximum information for identification and eliminating redundant electrodes with maximum accuracy is presented in [53]. The study utilized the Physiontet BCI dataset, which was also used in the present study, consisting of recordings from 109 subjects and 64 EEG acquisition channels, as well as a self-collected dataset comprising recordings from 30 individuals and 14 channels. The study focused on the acquisition protocols of eyes open (EC), eyes closed (EO), and relaxation and concentration. The application of the genetic algorithm resulted in a reduction in the number of electrodes to approximately 9 to 12 channels. Using the beta frequency band to train a Fine Gaussian kernel SVM classifier, the accuracy rates ranged from 94% to 98.9%. The selected electrodes were located in the frontal, central, and parietal lobes. These regions of the superficial cerebral cortex, which have been demonstrated to be crucial in biometric identification of individuals, coincide with those identified by the proposed method. However, our method yields slightly better identification results with the use of 9 to 12 channels than the proposed in [53], achieving an accuracy rate of up to 99.4%.

Meanwhile, in [54], the non-dominated sorting genetic algorithm (NSGA) was utilized to address the multi-objective optimization problem of decreasing the number of EEG channels while maximizing the accuracy of multi-class classification, increasing the number of accepted subjects with access, and maximizing the number of intruders rejected. The authors tested their method on a dataset composed of event-related potentials (ERPs) recorded from 26 subjects using 56 EEG channels. The authors extracted features related to signal energy and fractal dimension using empirical mode decomposition (EMD) for each channel. By employing the NSGA algorithm, they were able to select seven channels that resulted in a subject identification accuracy of up to 98% when using an RBF-SVM classifier. Compared to our proposed method in the present study, as shown in the Section 3, the method was able to achieve better accuracy results and a higher proportion of reduction in the number of electrodes used when using the same classifier (RBF-SVM). It should be noted, however, that the EEG signal acquisition strategies studied in relation to mental tasks were different. Regarding feature extraction, one of the most crucial stages in classification procedures, our method not only used information from each electrode separately on the EEG signal, but also extracted information about the interrelation between them through functional connectivity, demonstrating that it is a feature that enhances biometric identification based on EEG.

Recently, in [19], the authors applied the GA to two separate datasets. The first dataset comprised EEG recordings from 21 volunteers using 19 EEG channels with audio-evoked responses as EEG recordings. In the second dataset, the authors initially selected 19 channels of interest with the motor action and motor imagery acquisition protocols. This dataset was the Physionet BCI dataset, consisting of 109 subjects. In both cases, coherence (COH) was used as the functional connectivity metric feature that yielded the best results, and a convolutional neural network (CNN) was employed as the classifier. In the 109 subjects’ dataset, the number of electrodes was reduced from 19 to 15, and the proposed method’s performance was only slightly affected, achieving 97.74% accuracy in the motor imagery protocol. In the dataset collected by the authors themselves, the number of electrodes was reduced from 19 to 11, achieving an accuracy rate of 99.56%. Compared to the results obtained using the database of 109 subjects and motor imagery mental tasks, it should be noted that similar identification accuracy and degree of reduction in the number of electrodes were achieved with the proposed method. However, the regions of the cerebral cortex where the selected electrodes are located differ. The results of the proposed method highlight the influence of the right parietal and temporal and left frontal lobes, while those achieved in [19] also include the occipital lobes.

One alternative approach to the previous literature on optimizing the number of EEG channels for biometric identification was proposed in [55]. The authors used a frequently occurring maximum power algorithm to achieve this goal on two databases, using resting-state acquisition strategy: the previously mentioned Physionet database and a self-collected dataset of 16 subjects. In both datasets, they achieved a reduction from 64 to 20 channels with an equal error rate (ERR) value of 0.0039. The selected 20 channels were predominantly located in the left hemisphere’s frontal, fronto-temporal, and fronto-central lobes. Although the acquisition strategy used in [55] (resting state) differs from the one used in the present study for the same database (motor imagery), the brain regions identified are similar to those obtained with the proposed method in this study. Moreover, the achieved trade-off between the minimum number of electrodes and the maximum possible identification accuracy was comparable between both methods.

In recent years, several studies have employed metaheuristic optimization algorithms, including the approach presented in [56]. In this work, the authors proposed a methodology based on the Flower Pollination Algorithm (FPA) and β-Hill Climbing optimizer, named FPA β-hc, to select the EEG channels that provide the most relevant information for biometric identification. Two techniques have been utilized to extract features from each individual EEG channel: Wavelet Transform and Auto-regressive (AR) models. The method was tested on the above mentioned Physionet dataset using EEG motor imagery as the acquisition protocol strategy. The results showed that the proposed method achieved 100% accuracy in identifying the 109 subjects by reducing the number of electrodes to 35. Regarding the degree of reduction in the number of electrodes and the achieved identification accuracy, it should be noted that both this method and the one proposed in the present study achieved a sufficient level of accuracy. However, the number of electrodes required in the proposed method was significantly lower (16 electrodes). Furthermore, the proposed method was successfully tested on a database with signals acquired from a low-cost EEG device, where it was highlighted that only four electrodes were needed to achieve acceptable identification results.

## 5. Conclusions

A comparative analysis of two dimensionality reduction techniques (PCA and Wilcoxon signed-rank test) using an automatic classification algorithm for the biometric identification of individuals based on their particular patterns of brain nerve activity has been shown. In this context, it has been demonstrated that the techniques applied are feasible for the study of the optimal number of EEG signal acquisition electrodes needed to obtain a sufficient degree of accuracy in classification. Furthermore, based on this information, it has been shown which electrode location areas provide the most relevant information to the system for the mental task that has been carried out to enable their identification. In this way, it has been determined that these areas are close to the frontotemporal areas when the subjects were performing the motor imagery mental task, although a greater concentration has been seen on the left or right side depending on the dimensional reduction technique applied.

Using the results obtained in both methods, it has been demonstrated that it is possible to maintain sufficient accuracy ratios by using at least four acquisition sensors with a low-cost EEG device.

## Figures and Tables

**Figure 1 sensors-23-04239-f001:**
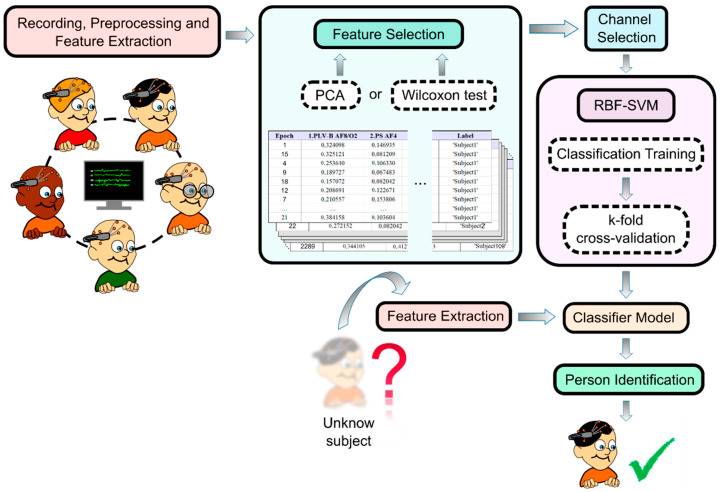
The steps followed for the identification of individuals using EEG-based biometric measurements.

**Figure 2 sensors-23-04239-f002:**
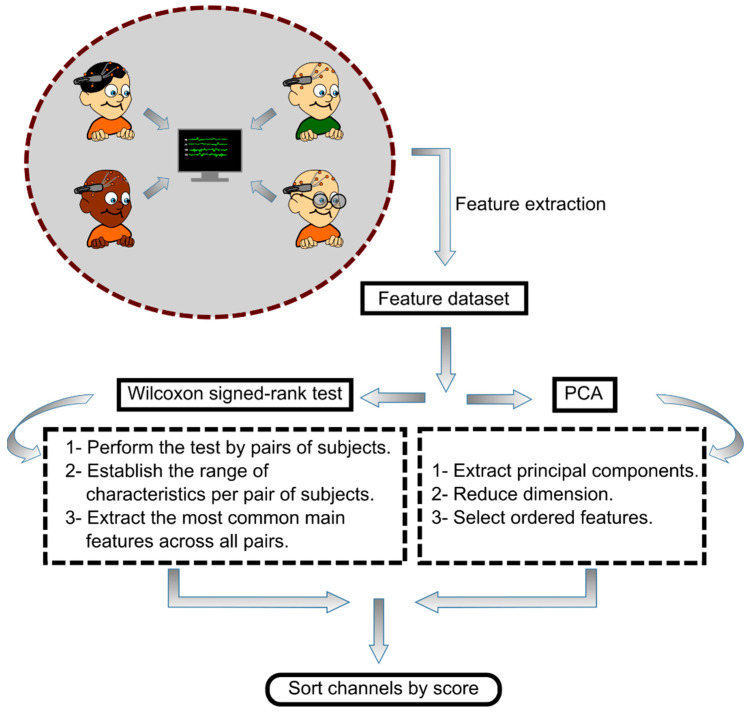
Flowchart of the EEG channel selection procedure using the Wilcoxon signed-rank test or PCA method.

**Figure 3 sensors-23-04239-f003:**
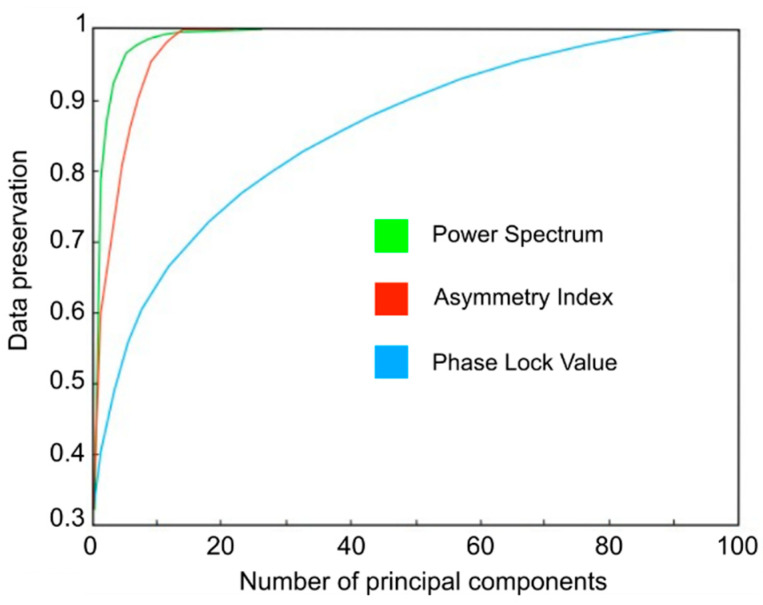
Ratio of conservation of the original data as a function of the number of principal components extracted for each group of characteristics by means of PCA.

**Figure 4 sensors-23-04239-f004:**
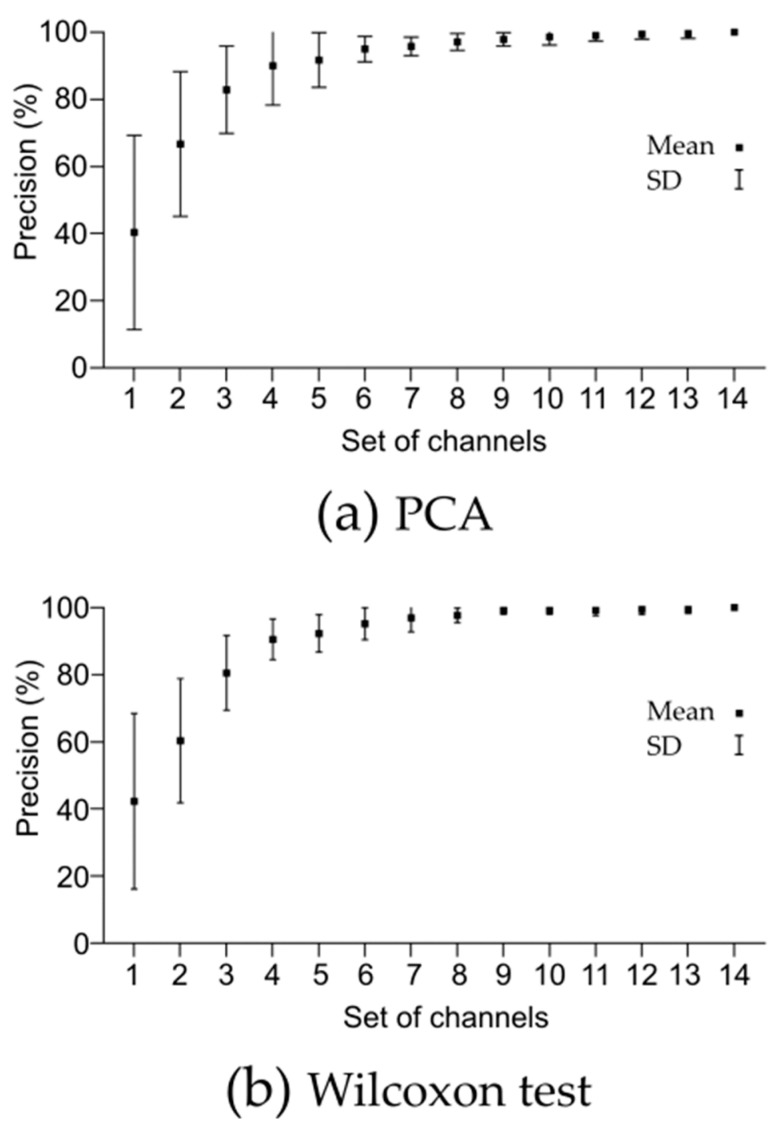
Precision obtained by applying PCA (**a**) and Wilcoxon signed-rank test (**b**) as a function of the number of acquisition channels.

**Figure 5 sensors-23-04239-f005:**
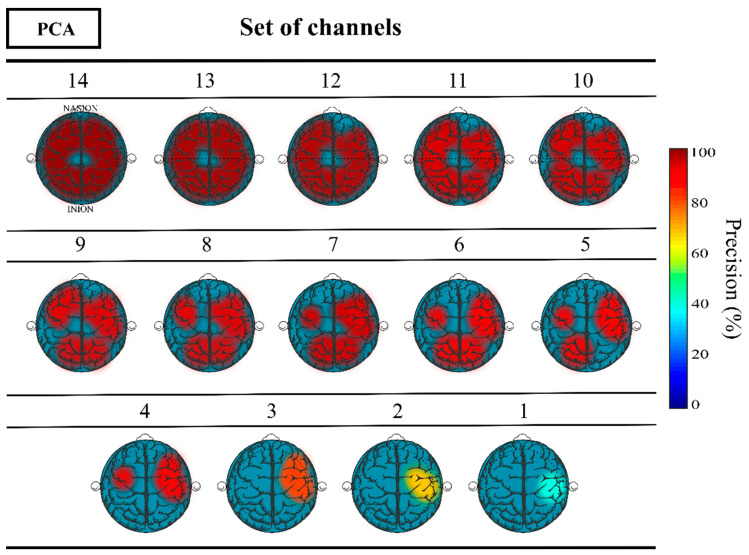
Topographic representation of the precision obtained as a function of the number of channels used and their corresponding location from the PCA application.

**Figure 6 sensors-23-04239-f006:**
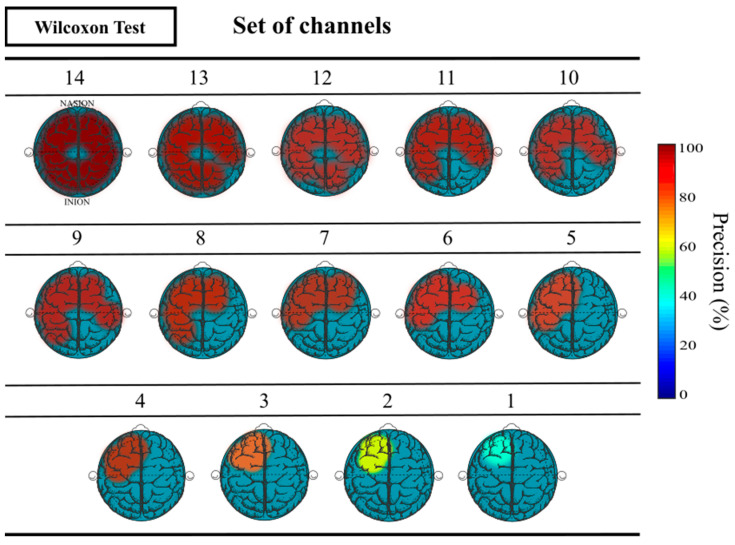
Topographic representation of the precision obtained as a function of the number of channels used and their corresponding location from the application of the Wilcoxon signed-rank test.

## Data Availability

Not applicable.

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
