# Peer review of "Selection of the Minimum Number of EEG Sensors to Guarantee Biometric Identification of Individuals"

_sensors, 2023, doi:10.3390/s23094239_

Round 1
Reviewer 1 Report
In this manuscript, the authors proposed a method for selecting and reducing the minimum number of EEG sensors to obtain effective biometric identification results. For the proposed method, three sets of features called power spectrum, asymmetry index, and phase-locking value were extracted from each EEG signal in the alpha and beta frequency bands.
The authors employed two dimensionality reduction techniques called PCA and Wilcoxon signed-rank test to find the optimal number of EEG signal acquisition electrodes for a sufficient classification accuracy rate.
The authors used a dataset containing EEG signals from thirteen volunteer healthy right-handed subjects in the experiments.
Their experimental results have shown that at least 4 EEG electrodes must be used to obtain an acceptable accuracy rate for biometric identification.
As a result, I think that the results presented in the manuscript are valuable for researchers working in EEG-based biometric identification. Therefore, it can be accepted after the major revisions noted below:
Major Revisions:
1) The results proposed in the manuscript must be compared with the other methods proposed in the literature, and the reference list should be improved in this sense.
2) The EEG dataset used in the experiments is very small. The dataset contains only thirteen people. I think it is not enough to support the results presented in the manuscript. As a result, the size of the dataset should be increased.
Author Response
Q1: “The results proposed in the manuscript must be compared with the other methods proposed in the literature, and the reference list should be improved in this sense.”
A1:
The results have been compared in the discussion section with recent literature methods that use similar databases and approaches to those of the present study. Some of these methods are as follows:
- Albasri, A.; Abdali-Mohammadi, F.; Fathi, A. EEG Electrode Selection for Person Identification Thru a Genetic-Algorithm Method. J Med Syst 2019, 43, doi:10.1007/s10916-019-1364-8.
- Moctezuma, L.A.; Molinas, M. Multi-Objective Optimization for EEG Channel Selection and Accurate Intruder Detection in an EEG-Based Subject Identification System. Sci Rep 2020, 10, 1–13, doi:10.1038/s41598-020-62712-6.
- Monsy, J.C.; Vinod, A.P. EEG‐based Biometric Identification Using Frequency‐weighted Power Feature. IET Biom 2020, 9, 251–258, doi:10.1049/iet-bmt.2019.0158.
- Ashenaei, R.; Asghar Beheshti, A.; Yousefi Rezaii, T. Stable EEG-Based Biometric System Using Functional Connectivity Based on Time-Frequency Features with Optimal Channels. Biomed Signal Process Control 2022, 77, 103790, doi:10.1016/j.bspc.2022.103790.
- Alyasseri, Z.A.A.; Alomari, O.A.; Papa, J.P.; Al-Betar, M.A.; Abdulkareem, K.H.; Mohammed, M.A.; Kadry, S.; Thinnukool, O.; Khuwuthyakorn, P. EEG Channel Selection Based User Identification via Improved Flower Pollination Algorithm. Sensors 2022, 22, 2092, doi:10.3390/s22062092.
Q2: “The EEG dataset used in the experiments is very small. The dataset contains only thirteen people. I think it is not enough to support the results presented in the manuscript. As a result, the size of the dataset should be increased.”
A2:
The reviewer claims that a dataset of 13 persons is too small. Taking this suggestion into consideration, we have included the performance results of the proposed method using the largest public database that has been used in the related literature, provided by the Physionet BCI and consisting of 109 people. Furthermore, the results obtained with this database of 109 subjects are comparable to those obtained through the self-collected database of 13 subjects. However, if one looks at the dataset sizes used in some of the most relevant and recent related literature about biometric people identification, their authors use even smaller or comparable datasets to the one used in the present study. For example, 10 subjects (Rahman, 2021), 5 subjects (Moctezuma), 7 subjects (Zeinali and Seyedarabi, 2019), 9 subjects (Kong, 2019), 16 subjects (Monsy 2020) and 12 subjects (Jayarathne 2020). Some of them have also used low-cost devices similar to the one used in this manuscript. Therefore, to our knowledge, we kindly consider these to be sufficient sample sizes to draw reliable conclusions based on the related literature.
○ Arafat Rahman, Muhammad E.H. Chowdhury, Amith Khandakar, Serkan Kiranyaz, Kh Shahriya Zaman, Mamun Bin Ibne Reaz, Mohammad Tariqul Islam, Maymouna Ezeddin, and Muhammad Abdul Kadir. Multimodal EEG and Keystroke Dynamics Based Biometric System Using Machine Learning Algorithms. IEEE Access, 9:94625–94643, jul 2021. ISSN 21693536. doi: https://doi.org/10.1109/ACCESS.2021.3092840.
○ Luis Alfredo Moctezuma, Alejandro A. Torres-García, Luis Villaseñor-Pineda, and Maya Carrillo. Subjects identification using EEG-recorded imagined speech. Expert Systems with Applications, 118:201–208, mar 2019. ISSN 09574174. doi: https://doi.org/10.1016/j.eswa.2018.10.004.
○ Mahsa Zeynali and Hadi Seyedarabi. EEG-based single-channel authentication systems with optimum electrode placement for different mental activities. Biomedical Journal, 42(4):261–267, aug 2019. ISSN 23194170. doi: https://doi.org/10.1016/j.bj.2019.03.005.
○ Wanzeng Kong, Luyun Wang, Sijia Xu, Fabio Babiloni, and Hang Chen. EEG Fingerprints: Phase Synchronization of EEG Signals as Biomarker for Subject Identification. IEEE Access, 7:121165–121173, 2019. ISSN 2169-3536. doi: https://doi.org/10.1109/ACCESS.2019.2931624.
○ Jijomon Chettuthara Monsy and Achutavarrier Prasad Vinod. EEG-based biometric identification using frequency-weighted power feature. IET Biometrics, 9(6):251–258, nov 2020. ISSN 2047-4938. doi: https://doi.org/10.1049/iet-bmt.2019.0158.
○ Isuru Jayarathne, Michael Cohen, and Senaka Amarakeerthi. Person identification from EEG using various machine learning techniques with inter-hemispheric amplitude ratio. PLOS ONE, 15(9): e0238872, sep 2020. ISSN 1932-6203. doi: https://doi.org/10.1371/journal.pone.0238872.
Reviewer 2 Report
The manuscript by Ortega et al. investigates which sensors out of a 14 sensor electrode setup contain the most information for identification of individuals based on their EEG signal during a motor task. The manuscript is well-written and the quality of the figures is sufficient. I would therefore recommend the manuscript for publication.
Author Response
We appreciate the comments provided by the reviewer.
Reviewer 3 Report
Dear authors,
The article offers the contribution in the field of EEG signals for biometric recognition, which makes it a valuable resource for researchers and practitioners in this field.
The article's methodology for selecting and reducing the minimum number of EEG sensors required for effective biometric identification is especially noteworthy.
The paper is well-written, but it would benefit from some revision. The abstract provides a clear and concise overview of the paper's main objectives and findings. However, there are a few areas where the writing could be improved.
The abstract could benefit from more detail on the methodology used to establish the minimum number of EEG sensors necessary for effective biometric identification. While the abstract briefly mentions the methodology, it does not provide enough detail for readers to fully understand the approach taken.
"The EEG has been widely used in medicine [11,12] and non- medical applications, such as in the development of brain–machine interfaces (BMIs) [13,14]. Unlike other biometric measurements, EEG-based biometric measurements of an individual are difficult to falsify, and it can be guaranteed that the identified person is alive."
I agree with the author's assertion that EEG signals are more difficult to falsify, but I would also like to suggest that the authors address some of the issues associated with EEG recordings. Specifically, I invite the authors to discuss how the variability and non-stationarity of EEG, as described in studies such as 10.1016/j.cmpb.2020.105808 and 10.1186/1753-4631-3-2, can introduce problems in biometric identification as it introduces in the classification of motor imageries, very close to the task described in your paper. For instance, changes in EEG features over time due to various reasons such as noise, drowsiness, and changes in electrode conditions can make it challenging to obtain the same EEG signal twice for the same person. To address this issue, proposals such as spatial filtering and stationarity subspace analysis (SSA) have been put forward to reduce non-stationarity, and the proper placement of electrodes (the topic of this paper) can also help eliminate problems related to feature covariance shifts. I believe that discussing these aspects in the introduction would enhance the paper's overall quality.
"...including two reference electrodes at P3 and P4) with saline-soaked felt pads. The electrical reference CMS (Common Mode Sense) is located at P3 (active electrode), and the reference DRL (driven right leg) is located at P4 (passive electrode)."
If I am not mistaken, only a single P3 electrode is used as the reference electrode, rather than two. The P4 electrode, as described later, is not a reference electrode. It is often referred to as the ground electrode and is necessary for enhancing the amplifier's common mode rejection ratio (CMRR) and signal-to-noise ratio.
"This procedure allowed us to obtain an overview of the impact of the selection of different frequency bands on EEG-based biometric identification that depended on the type of mental task needed."
The meaning of the sentence is ambiguous. Are you referring to the alpha and beta bands only, or are other bands also involved?
"Principal component analysis"
I am unclear on where the PCA was applied. While there is a definition of PCA, it is not clear what data it was used on. If PCA was applied to the features prior to classification, it should be clearly described before the SVM classification is discussed.
" Wilcoxon signed-rank test"
As previously mentioned for PCA, while the signed-rank test is defined in the paragraph, it is not clear how it was applied on your data. It is unclear what was being compared using the signed-rank test.
"Both the principal component analysis and the c test are statistical tests that allow the discovery and ordering of those variable characteristics of a data set that provide more significant information"
I would not say that the PCA is a statistical test and it is very different from the signed-rank test. The PCA can be described as one of the most broadly used statistical methods for the ordination and dimensionality-reduction of multivariate datasets, but I would not say it is a statistical test.
"For this reason, these two techniques are widely used in problems of dimensionality reduction of multivariate models through the selection of characteristics to simplify their complexity."
True for PCA, but not for the Wilcoxon signed-rank test. It is not clear how the Wilcoxon signed-rank test can be used for the dimensionality reduction of multivariate datasets.
"In the application of biometric people identification by EEG proposed in this study, the aforementioned methods – PCA and Wilcoxon signed-rank test – have been used to reduce the number of EEG acquisition channels and to obtain the optimum required to achieve a desired accuracy."
This is the answer for some of the previous questions and should be part of the methods section, before classification (feature selection).
"n this way, the superficial regions of the cerebral cortex corresponding to the location of these favourite electrodes located on the scalp (AF3, F7, F3, FC5, T7, P7, O1, O2, P8, T8, FC6, F4, F8 and AF4) were ordered according to the relevance contained in the information provided by each one."
There is no need to repeat already described in the methods section.
I recommend that the authors separate the Results and Discussion sections. Currently, there is a lot of repetition of the methodology throughout the results, with little discussion of the findings. I suggest that the results section be more concise and that the applied methodology be moved to the methods section, focusing on how it was used rather than providing general descriptions of PCA and the signed-rank test.
1. What is the main question addressed by the research?
The main question addressed by the research is to establish a methodology for selecting and reducing the minimum number of EEG sensors necessary to carry out effective biometric identification of individuals.
2. Do you consider the topic original or relevant in the field? Does it
address a specific gap in the field?
The topic is both original and relevant in the field as it addresses the use of EEG signals for biometric recognition, which has recently been demonstrated. The article also addresses the specific gap in the field of how to select the minimum number of EEG sensors necessary for effective biometric identification.
3. What does it add to the subject area compared with other published
material?
The article adds to the subject area by providing a methodology for selecting the minimum number of EEG sensors necessary for effective biometric identification. The methodology can be used to reduce the cost and complexity of EEG-based biometric systems.
4. What specific improvements should the authors consider regarding the
methodology? What further controls should be considered? (CRITICAL)
The authors could consider improving the methodology. As I wrote in the original comments attached below:
- Introduce how nonstationarity influences biometric identification for example, by examining the accuracy of recording EEG with a specific number of channels using the proposed methods at two different times with a 30-minute gap. This analysis will help determine the approach's ability to handle nonstationarity in EEG signals.
- The descriptions of the reference electrode(s) contain inconsistencies and errors.
- I am unclear on where the PCA was applied. While there is a definition of PCA, it is not clear what data it was used on. If PCA was applied to the features prior to classification, it should be clearly described before the SVM classification is discussed.
- The authors incorrectly describe PCA as a statistical test similar to the signed-rank test. PCA is not a statistical test and is fundamentally different from the signed-rank test. Moreover, it is unclear what data the authors applied the signed-rank test to, necessitating clarification.
5. Are the conclusions consistent with the evidence and arguments presented
and do they address the main question posed? (CRITICAL)
To be improved. I think it would be best for the authors to separate the Results and Discussion sections. Currently, there is a lot of repetition of the methodology throughout the results, with little discussion of the findings.
6. Are the references appropriate?
The references are appropriate and cover relevant studies in the field.
7. Please include any additional comments on the tables and figures.
No additional comments on the tables and figures.
I recommend conducting a major review of the article to ensure that all sections are clear, and well-organized, and effectively communicate the study's objectives, methodology, findings, and conclusions. This review should include addressing any inconsistencies or errors in the descriptions of the reference electrodes, accurately characterizing statistical tests such as PCA and the signed-rank test, and providing more details about the data used in the study.
Author Response
Q1: Introduce how nonstationarity influences biometric identification for example, by examining the accuracy of recording EEG with a specific number of channels using the proposed methods at two different times with a 30-minute gap. This analysis will help determine the approach's ability to handle nonstationarity in EEG signals.
A1:
Following the suggestions and recommendations of the reviewer, the introduction section has been modified to describe and consider the complexities associated with the acquisition of electroencephalography signals related to the variability and non-stationarity of EEG. Additionally, the suggested bibliographic references related to this topic have been added.
Q2: The descriptions of the reference electrode(s) contain inconsistencies and errors.
A2: Regarding the reviewer's consideration, technical terms related to the device's electronic configuration were redefined in a clearer way to improve understanding as below:
“The electrical reference point CMS (Common Mode Sense) is located at P3 or right mastoid (active electrode), and the noise cancellation electrode DRL (driven right leg) is located at P4 or left mastoid (passive electrode).”
This description follows the technical information given by Emotiv on its official web pages:
https://www.emotiv.com/knowledge-base/are-cms-drl-references-positioned-as-usually-around-the-mastoid-are-p3-and-p4-signals-estimated-from-data-of-existing-closest-derivations/
https://www.emotiv.com/knowledge-base/everything-you-need-to-know-about-reference-sensors-of-emotiv-hardware/
Q3: I am unclear on where the PCA was applied. While there is a definition of PCA, it is not clear what data it was used on. If PCA was applied to the features prior to classification, it should be clearly described before the SVM classification is discussed.
A3: Considering the reviewer's suggestion, an extensive reorganization of the manuscript has been carried out, and more detailed information has been provided to offer an extended explanation and facilitate understanding of the proposed feature extraction techniques employed and channel selection method (based on PCA or Wilcoxon signed-rank test) prior to defining the classification stage.
Q4: The authors incorrectly describe PCA as a statistical test similar to the signed-rank test. PCA is not a statistical test and is fundamentally different from the signed-rank test. Moreover, it is unclear what data the authors applied the signed-rank test to, necessitating clarification.”
A4: Following the suggestions and recommendations of the reviewer, a substantial reorganization of the manuscript was conducted and we presented more comprehensive details to provide an extended explanation and enhance comprehension of the PCA and its formal definition was corrected. The application procedure of the Wilcoxon signed-rank test and the data involved have been clarified for a better understanding. In addition, the procedure to use the Wilcoxon signed-rank test for the dimensionality reduction and of multivariate datasets channel selection was clarified in the methods section as follows:
“When using the Wilcoxon signed-rank test to sort the channels by the p-value of their characteristics involved, a reduced array of selected features was extracted. Unlike the PCA method, the Wilcoxon test can be performed only on paired data sets; that is, between the data of only two people at a time. For this reason, it was necessary to perform the test on 77 combinations of pairs of subjects for 13 subjects. Considering all the tests that sorted the features according to their p-value and the corresponding group of features, they were reordered by their statistical mode or the number of times they were repeated in the features array ordered by the Wilcoxon test among all the tests”.
Q5: To be improved. I think it would be best for the authors to separate the Results and Discussion sections. Currently, there is a lot of repetition of the methodology throughout the results, with little discussion of the findings.
I recommend conducting a major review of the article to ensure that all sections are clear, and well-organized, and effectively communicate the study's objectives, methodology, findings, and conclusions. This review should include addressing any inconsistencies or errors in the descriptions of the reference electrodes, accurately characterizing statistical tests such as PCA and the signed-rank test, and providing more details about the data used in the study.
A5: In accordance with the reviewer's feedback, a significant and comprehensive restructuring of the manuscript has been undertaken, incorporating distinct sections for results and discussion to enhance comprehension and argument clarity. Moreover, a comparative analysis between the proposed methods outlined in the current manuscript and relevant recent literature has been expanded within the discussion.
Reviewer 4 Report
The article is very interesting and innovative, but it needs corrections. In the Material and Methods section, the numbering should be corrected. It should be separate to write the results and the discussion separately, because it is difficult to read.
Author Response
Q1: The article is very interesting and innovative, but it needs corrections. In the Material and Methods section, the numbering should be corrected. It should be separate to write the results and the discussion separately, because it is difficult to read.
A1: The numbering in the Materials and Methods section has been corrected. Considering the reviewer’s suggestions, a substantial and extensive reorganization of the manuscript has been carried out, including separate sections dedicated to the results and discussion for better understanding and clarity of the arguments presented. The introduction section was improved to provide sufficient background including all relevant references. The methods description section has been carefully improved. Additionally, the discussion has been extended through a comparison between the proposed methods in the present manuscript and related recent literature.
Reviewer 5 Report
1. This paper needs to add descriptions to the experimental process, such as the use of test set , training set and input sample size.
2. More details need to be added to the description of the experimental procedure Figure (Figure 1), such as classification training block, verification block et al.
3. Why use phase locking value? What are the advantages compared with other feature extraction methods?
4. How long does the Wilcoxon signed-rank test method run on what computer configuration?
5. Compared with the methods of deep learning, what are the advantages of the classification method proposed in this paper?
Author Response
Q1: This paper needs to add descriptions to the experimental process, such as the use of test set , training set and input sample size.
A1: The feature extraction and the classification section have been modified to include a more detailed explanation of the use of the test set, training set and input sample size. The following explanation has been included in the manuscript:
2.4. Feature extraction
“...Hence, in the case of dataset I, every subject was represented by 112 features for every epoch in the beta frequency band. These features included 14 PS features, 7 AI features, and 91 PLV features. Similarly, for dataset II, each subject was characterized by 2107 features from the beta frequency band, comprising of 64 PS features, 27 AI features and 2016 PLV features. The dimension of the feature tables, which includes the subject's label column, was 520x112 and 2289x2108 for dataset I and II, respectively, when using all available electrodes.”
- Classification
“...The feature tables extracted from EEG dataset I and II, which contained the PS, AI, and PLV features measured in the beta frequency band, were split into training and validation sets. This split was performed using a 10-fold cross-validation technique, which is commonly used in machine learning to evaluate the performance of a model while preventing overfitting. In cross-validation, the data is divided into k equal parts, and the model is trained k times, with each part serving as the validation set once. The average performance across all the k-folds is then reported as the overall performance of the model. This approach helps to ensure that the model generalizes well to unseen data and is not overly influenced by noise or outliers in the training set. In this way, the inputs of the classifier are the feature data and the output is the label of the corresponding subject.”
Q2: More details need to be added to the description of the experimental procedure Figure (Figure 1), such as classification training block, verification block et al.
A2: The experimental procedure Figure (Figure 1) has been carefully improved considering the reviewer’s suggestions.
Q3: Why use phase locking value? What are the advantages compared with other feature extraction methods?
A3: Phase lock value (PLV) relates the allocation or variability of the relative phase of different signals. This type of synchronous phase index describes the absolute value of the mean phase difference between the two signals and can identify transient phase-locking value independently of the signal amplitude. In the discussion section of the manuscript, supported by the results section, we show that considering the functional connectivity between the different EEG channels can provide reliable information for biometric identification:
“...Regarding feature extraction, one of the most crucial stages in classification procedures, our method not only used information from each electrode separately on the EEG signal, but also extracted information about the interrelation between them through functional connectivity (characterized by the PLV feature), demonstrating that it is a feature that enhances biometric identification based on EEG.”
In addition, we extended the introduction section to provide a more robust background regarding to the use of functional connectivity in the EEG-based biometric identification application:
“...Concerning the selection of features that improve the performance of biometric identification systems, studies have recently been published suggesting that the subtraction of information related to functional connectivity between different regions of the brain is a feature that can potentially improve pattern classification tasks in EEG signals. Several studies have demonstrated that features derived from functional connectivity can significantly enhance the performance of EEG signal classification for biometric recognition systems [18,19]. Incorporating such features can improve the robustness of biometric recognition systems by considering the interdependence of EEG channels [20]. One of the most effective methods for achieving this objective includes the study of the phase synchronization [21,22].”
Q4: How long does the Wilcoxon signed-rank test method run on what computer configuration?
A4: We have added this information to the manuscript. The computation running time for each feature extraction method was included. The computation time for the PCA method was 36.2 ms. The computation time for the Wilcoxon signed-rank test method was 7.37 ms. Regarding the computer configuration, it was indicated as follows:
2.6 Computation setup
“The calculations involved in the present study were performed on a computer with an AMD Ryzen 7 3800X processor with 8 cores and 16 threads at 4.5 GHz, an Nvidia RTX 2060 graphics card with 6 Gb of memory at 1.7 GHz, and four 4x16 Gb (64 Gb) RAM modules with a CAS latency of 16 at 3.2 GHz.”
Q5: Compared with the methods of deep learning, what are the advantages of the classification method proposed in this paper?
A5:
In the present work, an attempt has been made to identify the EEG electrodes and/or regions of the cerebral cortex that are essential to be able to use EEG signals in biometric identification systems. This would allow simplifying the design of an EEG system for such a purpose. In this work, it is not intended to compare this methodology with deep learning methods. On the other hand, one advantage of supervised classification algorithms, such as Support Vector Machines (SVM), over deep learning methods, is that the first ones can perform well with limited data, whereas deep learning often requires large amounts of data for training. Additionally, SVM models are generally easier to interpret and less computationally intensive compared to deep learning models.
Round 2
Reviewer 1 Report
The authors completed the requested corrections and additions. The manuscript can be accepted for publication as it is.
Reviewer 3 Report
Overall, the authors have addressed all the questions and suggestions.
A1:
The authors have addressed the reviewer's suggestion by modifying the introduction section to include a more detailed description of the complexities associated with EEG signal acquisition, particularly related to the non-stationarity and variability of EEG. They have also added relevant bibliographic references to support this description.
A2:
The terminology used may seem unconventional, but it is acceptable as it has been taken from the EMOTIV technical documentation.
A3:
The authors have reorganized the manuscript and provided more detailed information on the feature extraction techniques employed, including the channel selection method based on PCA or Wilcoxon signed-rank test. They have clarified the application of PCA and the specific data it was used on, prior to discussing SVM classification.
A4:
The authors have made substantial revisions to the manuscript, including a correction to the formal definition of PCA and clarification on the application procedure of the Wilcoxon signed-rank test. They have also provided more information on the specific data used for the Wilcoxon signed-rank test, which was performed on paired data sets of two people at a time. Furthermore, they have described the process of channel selection and feature extraction based on the results of the Wilcoxon test.
A5:
The authors have made significant revisions to the manuscript, including a restructuring of the Results and Discussion sections for greater clarity and organization. They have also provided more detailed analyses and comparisons of the proposed methods and their findings in relation to recent literature. Additionally, they have addressed inconsistencies and errors in the descriptions of technical terms and statistical tests, and provided more information on the data used in the study.
Reviewer 4 Report
The text is corrected as recommended. All information is provided in a clear way